# Monitoring cognitive resilience in military personnel in extreme operational environments: The role of smart technologies and nutritional strategies—A scoping review protocol

**Rodica Siminiuc**◉*, **Dinu Turcanu**◉

Department of Food and Nutrition, Technical University of Moldova, Chisinau, Republic of Moldova

* rodica.siminiuc@adm.utm.md

## Abstract

### Introduction

Maintaining cognitive performance in extreme operational environments is essential for military personnel efficiency. Exposure to multiple stressors, including sleep deprivation, intense physical exertion, and limited access to food resources, can affect cognitive function and decision-making ability. Although studies on military nutrition and smart technologies exist, an integrated synthesis that analyzes their interactions is currently lacking.

### Materials and methods

This scoping review follows the JBI methodology, with a structured study selection and data synthesis process. The analysis includes international databases and grey literature, with results presented both narratively and in tabular form,

### Objectives

This scoping review aims to map and analyze the literature on smart technologies used for monitoring the cognitive resilience of military personnel deployed in extreme operational environments, as well as the role of nutritional strategies in supporting cognitive resilience, following the PCC criteria.

### Inclusion criteria

Studies published after 2000 that investigated nutritional strategies and the use of smart technologies to monitor or support the cognitive resilience of military personnel will be included. Research focusing solely on physical performance or pharmacological interventions unrelated to operational nutrition was excluded,

**Data availability statement:** No datasets were generated or analysed during the current study. This is a scoping review protocol. De-identified data resulting from the review will be made publicly available in an open-access repository (e.g., OSF) upon completion of data extraction and synthesis.

**Funding:** This study was supported by the National Agency for Research and Development (NARD), Republic of Moldova, in the form of a grant awarded to D.T. (25.00208.707.05.PD). The specific roles of this author are articulated in the 'Author Contributions' section. The funders had no role in study design, data collection and analysis, decision to publish, or preparation of the manuscript.

**Competing interests:** The authors have declared that no competing interests exist.

## Registration

This scoping review protocol was registered in the OSF (Open Science Framework): [https://doi.org/10.17605/OSF.IO/EC7MG].

## Introduction

Adaptive military operations are a defining feature of modern warfare that rely on the ability of military personnel to demonstrate flexible and resilient behavior. Service members are subjected to extreme physical and mental demands, requiring the capacity to overcome the negative effects of physiological and psychological stress imposed by challenging operational environments characterized by "volatile, uncertain, complex, and ambiguous" events [1].

A key objective of nutritional science research is to develop and validate nutritional interventions tailored to military personnel, ensuring optimal support for both physical and cognitive health. Studies indicate a growing interest within the United States military in investigating the impact of nutritional factors on soldiers' resilience, considering both physical and cognitive performance perspectives [2,3].

Research suggests that diets rich in monounsaturated fatty acids, polyphenols, and essential vitamins contribute to improved vascular health and white matter microstructure, thereby supporting cognitive function and mental adaptability [4,5]. Additionally, adequate carbohydrate intake is essential for maintaining the optimal glucose levels required for brain function under high-stress conditions. Furthermore, while nutritional supplements are commonly used by military personnel, their efficacy remains inconclusive due to limited support from randomized controlled trials [6,7].

In recent years, smart and emerging technologies have been explored as innovative solutions to optimize the nutritional status of military personnel. Wearable biosensors, artificial intelligence, and digital metabolic analysis platforms enable real-time monitoring of physiological markers and individual nutritional needs. However, the application of these technologies in military nutrition remains under-documented and requires further research for effective integration into operational strategies [2,8].

### Definition of key concepts.

To clarify the conceptual framework of this review, the following key terms are defined:

*Cognitive resilience* refers to the ability to maintain or recover cognitive function despite exposure to physiological, pathological, or operational stressors, enabling adaptation and optimal performance under adverse conditions [9,10].

### *Nutritional strategies*:

Structured dietary interventions designed to support and monitor the metabolic and neurocognitive status of military personnel by adjusting macronutrients, micronutrients, and fluid intake according to operational demands and environmental stressors [3,8].

*Smart technologies* are advanced devices and algorithms (e.g., biosensors, artificial intelligence, predictive analytics) that enable real-time monitoring of physiological and cognitive parameters relevant to nutritional status and cognitive resilience in extreme operational environments [2,3,8].

*Extreme operational environment* – Military conditions characterized by exposure to extreme temperatures, intense physical exertion, sleep deprivation, and limited access to nutritional resources [11,12].

### Rationale.

Although specialized literature includes studies on military nutrition [13–15], cognitive performance in extreme environments, and the use of smart technologies in nutrition [16,17], there is no systematic analysis that integrates these dimensions into an operationally applicable model. Existing studies either examined the impact of nutrition on cognitive performance or the role of emerging technologies in monitoring nutritional status, but they did not explore the interaction between these variables in a military context. This lack of integration makes it difficult to develop evidence-based interventions that support both cognitive resilience and rapid decision making in extreme operational environments. A scoping review is necessary to synthesize existing literature, identify research gaps, and establish a foundation for more in-depth future studies.

### Preliminary search and justification for a scoping review.

A preliminary search in PubMed, Cochrane Database of Systematic Reviews, and JBI Evidence Synthesis did not identify any systematic or scoping reviews that comprehensively investigated the integrated relationship between nutrition, smart technologies, and cognitive resilience in extreme operational contexts. Existing studies address these elements separately, either by analyzing the impact of nutrition on cognitive performance or by examining the use of emerging technologies for monitoring nutritional status.

This scoping review is justified, as it provides the first systematic synthesis of the literature on this topic, mapping current trends, research gaps, and future opportunities for investigation.

### Objective.

This scoping review aims to map and analyze the scientific literature on nutritional strategies and smart technologies applicable to monitoring and supporting the cognitive resilience of military personnel deployed in extreme operational environments, identifying current trends, research gaps, and opportunities for practical implementation.

### Review question

This scoping review will aim to answer the following questions using the PCC framework (Table 1):

The scoping review will address these questions without testing a priori hypotheses; the goal is to comprehensively map the evidence base and identify knowledge gaps.

### Inclusion criteria

*Population (P)*:   This review includes studies investigating nutrition, smart, and emerging technologies in relation to the monitoring and support of cognitive resilience in military personnel. Only research conducted in operational contexts relevant to extreme environments specific to the armed forces will be included.

**Concept (C)**:   Studies examining nutritional strategies and smart technologies used to monitor and support cognitive resilience in extreme operational environments will be included. The analysis will cover nutritional interventions (e.g., supplementation, combat rations, and personalized diets) and the use of emerging technologies—defined as wearable biosensors, digital platforms, or AI-driven systems that monitor physiological or cognitive markers in real time without

**Table 1. Research questions formulated according to the PCC (Population, Concept, Context) framework.**

| Component | Research Question |
|---|---|
| P-Population | What categories of military personnel are investigated in the scientific literature regarding the impact of nutrition and smart technologies on cognitive resilience in extreme operational environments? |
| C-Concept | What nutritional strategies and smart technologies are documented for monitoring and supporting the cognitive resilience of military personnel? |
| C-Context | What factors influence and what challenges are associated with the use of personalized nutrition and smart technologies in extreme operational environments? |

intervening to alter dietary intake during the study period—for tracking physiological and cognitive responses to nutrition. In this context, the term 'personalized diets' refers to physiologically adapted nutrition strategies based on individual energy needs, mission type, and duration of operational stress exposure. This definition excludes subjective preferences, considering the logistical limitations and operational standardization requirements that restrict the feasibility of fully individualized nutrition in military contexts.

**Context (C):** Studies investigating the influencing factors, barriers, and opportunities related to the implementation of nutritional strategies and smart technologies in military nutrition will be included. This review will analyze research on operational constraints, access to nutritional resources, and the impact of extreme conditions (e.g., extreme temperatures, sleep deprivation, and high stress) on the feasibility and effectiveness of these strategies.

## Types of sources

This scoping review will include primary studies (randomized controlled trials, cohort studies, case-control studies, cross-sectional studies, and observational studies), qualitative studies (phenomenology, ethnography, grounded theory), and mixed-methods studies. Additionally, systematic and scoping reviews relevant to the topic will be included, along with grey literature, technical reports, military guidelines, and government documents related to nutrition and smart technologies in extreme military contexts.

**Timeframe:** Studies published between 2000 and the present are included to reflect the recent evolution of research in this field.

**Language:** Publications in English, French, Spanish, German, Romanian, and Russian will be analyzed. Articles in English, French, Romanian, and Russian will be assessed directly by fluent co-authors. For Spanish and German publications, initial screening and data extraction will be supported by AI-assisted translation tools, with further verification by a second reviewer or language specialist as needed.

**Exclusion criteria:** Studies that did not investigate military personnel or analyze nutrition without a clear connection to cognitive resilience in extreme operational environments were excluded. Studies investigating dietary supplements, ergogenic aids, or pharmacological compounds will be excluded unless the intervention is explicitly formulated for, evaluated in, or operationally relevant to active-duty military personnel during training or deployment. Consequently, trials conducted exclusively in sedentary, clinical, or general civilian cohorts—without a direct military performance context—will not be considered.

The proposed scoping review will be conducted following the JBI methodology for scoping reviews, using the JBI Manual for Evidence Synthesis [18] and reported in accordance with PRISMA-ScR (Preferred Reporting Items for Systematic Reviews and Meta-Analyses Extension for Scoping Reviews) [19,20]. This protocol has been registered at OSF [https://doi.org/10.17605/OSF.IO/EC7MG]. At the time of submission, this scoping review is in the initial data extraction phase. Upon acceptance of this protocol, record screening will commence immediately and is projected to be completed within eight weeks. Data extraction will then follow and is anticipated to take six weeks. Results synthesis and manuscript

drafting are scheduled to be completed within eight weeks thereafter. Any subsequent timeline adjustments will be documented in the final publication. A detailed log of modifications will be maintained to ensure transparency.

## Search strategy

The search strategy included both published sources and grey literature, following a three-stage approach:

Stage 1: An initial exploratory search of MEDLINE (PubMed) was conducted to identify relevant keywords and optimize the query strategy.

Stage 2: A comprehensive search of major databases, including PubMed/MEDLINE, Web of Science, Scopus, and IEEE Xplore, using keywords and indexed terms adapted to each database.

Stage 3: A grey literature review incorporating sources such as CyberLeninka, eLIBRARY.ru, and relevant government reports (NATO, WHO, DARPA) to complement the academic literature.

The full list of databases and an example PubMed query will be included in Appendix I. The search strategies for the other databases will be documented and reported in the final publication.

## Study selection

All identified citations were imported and managed using Zotero [6.0.37] (Corporation for Digital Scholarship, VA, USA), with duplicates removed both automatically and manually. The screening process was conducted in two stages by two independent reviewers using Rayyan software. In the first stage, titles and abstracts were screened based on inclusion/exclusion criteria. In the second stage, full-text reviews were conducted for all relevant articles. Any discrepancies between the reviewers were resolved through discussion or, if necessary, by involving a third reviewer.

The study selection process will be documented using the PRISMA Flow Diagram, providing a transparent overview of the number of records identified, screened, included, and excluded at each stage [19–21]. Reasons for study exclusion at the full-text level will be documented in an appendix to enhance transparency.

## Data extraction

Data were extracted using a standardized extraction tool developed in accordance with the JBI Manual for Evidence Synthesis [18]. The extracted variables will include the investigated population, referring to the categories of military personnel analyzed in the studies; documented nutritional strategies that describe the dietary interventions used to support cognitive resilience in military personnel; utilized smart and emerging technologies, including devices and technological solutions applied in operational nutrition; and the operational context, detailing the extreme environments and conditions in which these strategies are implemented. Additionally, the year of publication and any references to military or institutional nutritional standards in place at the time of the study will be extracted, to allow interpretation of the interventions within their historical and regulatory context.

Furthermore, the data extraction process will account for the operational profile of the military personnel included in each study, such as the type of mission (e.g., long-duration, low-intensity vs. short-duration, high-risk missions) and the specific military specialization (e.g., combat troops, special forces, logistics, aviation). This distinction is critical for contextualizing nutritional strategies and interpreting the applicability and relevance of monitoring technologies, as cognitive and physiological demands vary significantly across operational roles.

To enhance the relevance and applicability of the findings, studies will be examined for the technical specifications and validation criteria of the monitoring technologies used (e.g., accuracy, reliability under field conditions, and level of invasiveness). Where available, information regarding the type of biomarkers assessed (e.g., physiological, cognitive, metabolic) and the integration of real-time data with behavioral or cognitive outcomes will be extracted and reported. This will support a more comprehensive analysis of how smart technologies contribute to the monitoring of nutritional interventions in operational settings.

Data extraction will be conducted independently by two reviewers and any discrepancies will be resolved by consensus. A draft version of the extraction tool was piloted and adjusted throughout the study to ensure reliability.

## Data analysis and presentation

Data will be synthesised narratively and in tabular form, following the PCC framework (Population, Concept, Context). All exploratory analyses, such as thematic mapping or subgroup comparisons (e.g., by service branch or environment), will be clearly labelled as exploratory. Any deviations from the planned methods will be recorded in a protocol deviation log and reported in the final manuscript.

Results will be presented as descriptive summaries supported by comparative tables and graphical visualisations, including thematic maps and diagrams that illustrate the geographic and temporal distribution of studies. A descriptive analysis of key findings will highlight current trends and remaining knowledge gaps in the literature.

## Supporting information

**S1 Appendix. Search strategy.**
(DOCX)

## Author contributions

**Conceptualization:** Rodica Siminiuc.

**Project administration:** Dinu Turcanu.

**Writing – original draft:** Rodica Siminiuc, Dinu Turcanu.

**Writing – review & editing:** Rodica Siminiuc, Dinu Turcanu.

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
