## [Decision Letter · Decision Letter 0]

PONE-D-25-09172Monitoring Cognitive Resilience in Military Personnel in Extreme Operational Environments: The Role of Smart Technologies and Nutritional Strategies – A Scoping Review ProtocolPLOS ONE

Dear Dr. Siminiuc,

Thank you for submitting your manuscript to PLOS ONE. After careful consideration, we feel that it has merit but does not fully meet PLOS ONE’s publication criteria as it currently stands. Therefore, we invite you to submit a revised version of the manuscript that addresses the points raised during the review process.

We look forward to receiving your revised manuscript.

Kind regards,

Rogis Baker, Ph.D

Academic Editor

PLOS ONE

2. We notice that your supplementary tables are included in the manuscript file. Please remove them and upload them with the file type 'Supporting Information'. Please ensure that each Supporting Information file has a legend listed in the manuscript after the references list.

Reviewers' comments:

Reviewer's Responses to Questions

**Comments to the Author**

1. Does the manuscript provide a valid rationale for the proposed study, with clearly identified and justified research questions?

Reviewer #1: Yes

Reviewer #2: Partly

2. Is the protocol technically sound and planned in a manner that will lead to a meaningful outcome and allow testing the stated hypotheses?

Reviewer #1: Partly

Reviewer #2: Partly

3. Is the methodology feasible and described in sufficient detail to allow the work to be replicable?

Reviewer #1: Yes

Reviewer #2: Yes

4. Have the authors described where all data underlying the findings will be made available when the study is complete?

Reviewer #1: Yes

Reviewer #2: Yes

5. Is the manuscript presented in an intelligible fashion and written in standard English?

Reviewer #1: Yes

Reviewer #2: Yes

6. Review Comments to the Author

You may also provide optional suggestions and comments to authors that they might find helpful in planning their study.

Reviewer #1: The article addresses an important and current topic. At the outset, it should be emphasized that the literature review methodology employed by the Authors has been presented clearly and appears to be formally correct. The method of analysis and synthesis of available sources has been described appropriately.

Nevertheless, for the purpose of further analysis and a fuller understanding of its implications, the following issues are worth considering, as they may be a subject for further discussion or supplementation:

1. Time scope of analyzed publications and evolution of nutritional standards.

The use of articles published over various years in the review is understandable from the perspective of a comprehensive approach to the topic. However, it should be noted that such a time span entails certain limitations. Dietary requirements and standards for other nutritional interventions (e.g., supplementation, design of combat rations, or implementation of personalized diets) have undergone significant changes over the years. Therefore, the question arises as to whether the Authors have considered these dynamic changes and potential restrictions concerning top-down imposed nutritional norms in different periods and armed forces. It is also crucial to specify what exactly is understood by the term "personalized diets" in the context of the analyzed studies – whether it refers to individualization based on biological markers, preferences, or other factors, especially in military realities where logistics and standardization play a significant role.

2. Operational specificity versus monitoring of cognitive resilience.

In the context of monitoring Cognitive Resilience in military personnel using smart technologies and nutrition, a crucial aspect appears to be the differentiation of analysis based on the type of operations performed and, consequently, different training categories and specializations of soldiers. Have the Authors considered how different mission profiles (e.g., long-duration, low-intensity operations vs. short-duration, high-risk, high-stress operations) and training specifics (e.g., pilots, special forces, logistics) might modulate both nutritional needs and responses to interventions, as well as the interpretation of data from monitoring systems? Different tasks and profiles can be associated with varying cognitive and physiological loads.

3. Criteria for selecting monitoring technologies and data integration.

Nutritional intervention is a complex process. Consequently, questions arise regarding the technical aspects of real-time monitoring. What are the detailed criteria for the selection and validation of monitoring equipment (e.g., accuracy, reliability in field conditions, minimal invasiveness)? Will the article include a review or recommendations for specific types of biomonitors (e.g., wearable sensors, sweat, blood, saliva analyzers) capable of tracking markers relevant from the perspective of nutritional status and cognitive functions? Furthermore, it is fundamental to explain how data from biomonitors, collected in real-time, can be effectively integrated and correlated with the assessment of the impact of nutritional interventions on observable behaviors, psychophysical performance, and cognitive resilience of soldiers in a dynamic operational environment.

Reviewer #2: In the "Funding" section of the manuscript body (lines 189-191), it clearly states: "This research was funded by the National Agency for Research and Development (NARD) through a postdoctoral grant...".

However, in the "Financial Disclosure" section under "Additional Information" at the beginning of the document, the response to "Did you receive funding for this work?" is: "The author(s) received no specific funding for this work."

These two statements regarding funding sources are completely contradictory. This is a serious issue that requires immediate clarification and correction. Please ensure consistency in the declaration across all sections.

"Free Full Text" Restriction: In Appendix I, under "Search Filters Applied," it specifies "Access Type: Free Full Text." This is a very significant limitation. The goal of a scoping review is to broadly collect and map all relevant literature in the field. Limiting the search to only free full-text articles will lead to the omission of a substantial amount of potentially relevant and important research that requires institutional subscriptions or payment, which could result in systematic bias and incompleteness of the review's findings. It is strongly recommended to remove this restriction and to explain how non-open access literature will be obtained (e.g., through institutional library resources).

Presentation of the PubMed Search Strategy: The PubMed search strategy presented in Appendix I (Table A1) appears more like a series of exploratory, separate sub-searches (e.g., searching separately for "military nutrition and cognitive resilience," "cognitive resilience in a military context," and "smart technologies in a military context") rather than a final, comprehensive search query integrating all core concepts (nutrition, smart technologies, cognitive resilience, military personnel). The protocol should more clearly explain how the results of these sub-searches will be combined and used, or provide a more integrated Boolean search string capable of retrieving all relevant concepts simultaneously.

The protocol mentions (lines 139-140): "Record screening is expected to be completed by 15.04.2025, followed by the finalization of data extraction by 30.05.2025." Considering today's date is May 29, 2025, the deadline for data extraction (May 30, 2025) is imminent. If this protocol was recently submitted or is currently under review, this timeline may be inaccurate or overly tight. It is advisable to confirm or update the timeline.

Handling of Non-English Literature:

The protocol plans to include literature in multiple languages (English, French, Spanish, German, Romanian, Russian), which is commendable as it helps broaden literature coverage. However, the protocol does not mention how non-English literature will be translated, or how consistency and accuracy in data extraction from these different language sources will be ensured during the data extraction process. A brief explanation of the relevant process or resources could be beneficial.

The exclusion criteria state: "research focusing on dietary supplements or pharmacological compounds that are not applicable to military operational nutrition will not be included." The judgment of what is "not applicable to military operational nutrition" might be somewhat subjective. It is recommended to provide screeners with more specific defining criteria or examples to ensure consistency in the screening process.

Minor Suggestions and Clarifications:

In the "Additional Information" section, under "Data Availability," it states: "N/A - No results are reported" and "No datasets were generated or analysed during the current study." However, the main text (line 138) mentions: "At the time of submission, this scoping review is in the initial data extraction phase." If data extraction has already begun, it could be stated that "no datasets have been finalized or fully analyzed at this stage." A more precise statement could be: "Data extraction is currently underway. No datasets have been finalized or fully analyzed at the time of this protocol submission. All relevant data from this study will be made available upon study completion."

Detail in the "Concept" Definition:

In the "Inclusion Criteria" under "Concept (C)," when describing smart technologies, it mentions "emerging technologies for real-time tracking of physiological and cognitive responses to nutrition, without direct dietary adjustment." This limiting condition "without direct dietary adjustment" requires all reviewers to have a consistent understanding and application during literature screening.

7. PLOS authors have the option to publish the peer review history of their article (what does this mean? ). If published, this will include your full peer review and any attached files.

**Do you want your identity to be public for this peer review?** For information about this choice, including consent withdrawal, please see our Privacy Policy .

Reviewer #1: No

Reviewer #2: No

---

## [Author Response · Author response to Decision Letter 1]

12 Jun 2025

Dear Academic Editor and Reviewers,

Thank you for your thoughtful and constructive review of our scoping review protocol titled “Monitoring Cognitive Resilience in Military Personnel in Extreme Operational Environments: The Role of Smart Technologies and Nutritional Strategies.” We greatly appreciate your time, effort, and expertise.

We have carefully addressed all comments and revised the manuscript accordingly. All changes made in response to Reviewer 1 are marked in blue, those made in response to Reviewer 2 are marked in green, and changes reflecting feedback from both reviewers are marked in purple. These color-coded highlights have been applied directly in the revised manuscript to facilitate transparency and ease of cross-reference.

For each reviewer comment, we provide a detailed response below, clearly specifying the section(s) of the manuscript where the corresponding changes were made. Page and line numbers were not indicated, given that all modifications are explicitly marked by color within the text.

We hope that the revised manuscript meets the expectations of the journal and satisfies all reviewer concerns. We remain grateful for your feedback and are open to any further clarifications or suggestions you may have.

With kind regards,

Rodica Siminiuc, on behalf of all co-authors.

Reviewer #1: Response

The article addresses an important and current topic. At the outset, it should be emphasized that the literature review methodology employed by the Authors has been presented clearly and appears to be formally correct. The method of analysis and synthesis of available sources has been described appropriately.

Nevertheless, for the purpose of further analysis and a fuller understanding of its implications, the following issues are worth considering, as they may be a subject for further discussion or supplementation:

1.1. Time scope of analyzed publications and evolution of nutritional standards.

The use of articles published over various years in the review is understandable from the perspective of a comprehensive approach to the topic. However, it should be noted that such a time span entails certain limitations. Dietary requirements and standards for other nutritional interventions (e.g., supplementation, design of combat rations, or implementation of personalized diets) have undergone significant changes over the years. Therefore, the question arises as to whether the Authors have considered these dynamic changes and potential restrictions concerning top- down imposed nutritional norms in different periods and armed forces. It is also crucial to specify what exactly is understood by the term "personalized diets" in the context of the analyzed studies – whether it refers to individualization based on biological markers, preferences, or other factors, especially in military realities where logistics and standardization play a significant role. Thank you for the valuable observation.

We have updated the protocol to specify that, during the data extraction process, the year of publication and the applicable military or institutional nutritional standards at the time each study was conducted will be considered.

We have also explicitly clarified the meaning of the term “personalized diets” in the military context, emphasizing that it refers to physiologically adapted nutritional strategies rather than individual preferences, given the logistical constraints and operational standardization specific to military settings.

These additions were made in the sections “Inclusion Criteria – Concept (C)” and “Data Extraction”, and are marked in blue for clarity.

1.2. Operational specificity versus monitoring of cognitive resilience.

In the context of monitoring Cognitive Resilience in military personnel using smart technologies and nutrition, a crucial aspect appears to be the differentiation of analysis based on the type of operations performed and, consequently, different training categories and specializations of soldiers. Have the Authors considered how different mission profiles (e.g., long-duration, low-intensity operations vs. short-duration, high-risk, high-stress operations) and training specifics (e.g., pilots, special forces, logistics) might modulate both nutritional needs and responses to interventions, as well as the interpretation of data from monitoring systems? Different tasks and profiles can be associated with varying cognitive and physiological loads.

The protocol was updated to specify that, during the data extraction process, the operational profile of the military personnel analyzed in each study will be considered, including the type of mission and military specialization:

“Furthermore, the data extraction process will account for the operational profile of the military personnel included in each study, such as the type of mission (e.g., long-duration, low-intensity vs. short-duration, high-risk missions) and the specific military specialization (e.g., combat troops, special forces, logistics, aviation). This distinction is critical for contextualizing nutritional strategies and interpreting the applicability and relevance of monitoring technologies, as cognitive and physiological demands vary significantly across operational roles.”

This addition was made in the “Data Extraction” section.

1.3. Criteria for selecting monitoring technologies and data integration.

Nutritional intervention is a complex process. Consequently, questions arise regarding the technical aspects of real-time monitoring. What are the detailed criteria for the selection and validation of monitoring equipment (e.g., accuracy, reliability in field conditions, minimal invasiveness)?

Will the article include a review or recommendations for specific types of biomonitors (e.g., wearable sensors, sweat, blood, saliva analyzers) capable of tracking markers relevant from the perspective of nutritional status and cognitive functions? Furthermore, it is fundamental to explain how data from biomonitors, collected in real-time, can be effectively integrated and correlated with the assessment of the impact of nutritional interventions on observable behaviors, psychophysical performance, and cognitive resilience of soldiers in a dynamic operational environment.

The protocol was updated to specify that, where available, data will be extracted on the technical specifications and validation criteria of the monitoring technologies, the types of biomarkers assessed, and how real-time data are integrated with cognitive or behavioral outcomes. This addition was made in the “Data Extraction” section.

Reviewer #2:

2.1. In the "Funding" section of the manuscript body (lines 189-191), it clearly states: "This research was funded by the National Agency for Research and Development (NARD) through a postdoctoral grant...".

However, in the "Financial Disclosure" section under "Additional Information" at the beginning of the document, the response to "Did you receive funding for this work?" is: "The author(s) received no specific funding for this work."

These two statements regarding funding sources are completely contradictory. This is a serious issue that requires immediate clarification and correction. Please ensure consistency in the declaration across all sections.

We have corrected the “Financial Disclosure” section to clarify that no funding was received for conducting the research; however, the publication fee will be covered by a postdoctoral grant provided by the National Agency for Research and Development (NARD). The funder had no role in the study design, data analysis, manuscript preparation, or decision to publish. The information is now consistent and aligned with the journal’s transparency requirements.

2.2. "Free Full Text" Restriction: In Appendix I, under "Search Filters Applied," it specifies "Access Type: Free Full Text."

This is a very significant limitation. The goal of a scoping review is to broadly collect and map all relevant literature in the field. Limiting the search to only free full-text articles will lead to the omission of a substantial amount of potentially relevant and important research that requires institutional subscriptions or payment, which could result in systematic bias and incompleteness of the review's findings. It is strongly recommended to remove this restriction and to explain how non-open access literature will be obtained (e.g., through institutional library resources).

Thank you for the justified observation.

We have corrected this error in Appendix I by removing the “Free Full Text” restriction. We have clearly stated that no access restrictions were applied and that articles behind paywalls will be accessed through institutional library resources or via interlibrary loan services, as appropriate. The modification is marked in green in Appendix I.

2.3. Presentation of the PubMed Search Strategy:

The PubMed search strategy presented in Appendix I (Table A1) appears more like a series of exploratory, separate sub-searches (e.g., searching separately for "military nutrition and cognitive resilience," "cognitive resilience in a military context," and "smart technologies in a military context") rather than a final, comprehensive search query integrating all core concepts (nutrition, smart technologies, cognitive resilience, military personnel). The protocol should more clearly explain how the results of these sub-searches will be combined and used, or provide a more integrated Boolean search string capable of retrieving all relevant concepts simultaneously.

We have clarified the structure of Appendix I.

Table A1 now includes: (i) the initial integrated search covering all four concepts (0 results); (ii) the revised integrated search (nutrition + resilience + military personnel), which was used for screening (16 results); and (iii) two additional thematic sub-searches designed to maximize sensitivity. We also added an explanation regarding the combination and deduplication of search results. These modifications are marked in green.

2.4. The protocol mentions (lines 139-140): "Record screening is expected to be completed by 15.04.2025, followed by the finalization of data extraction by 30.05.2025."

Considering today's date is May 29, 2025, the deadline for data extraction (May 30, 2025) is imminent. If this protocol was recently submitted or is currently under review, this timeline may be inaccurate or overly tight. It is advisable to confirm or update the timeline.

Handling of Non-English Literature:

The protocol plans to include literature in multiple languages (English, French, Spanish, German, Romanian, Russian), which is commendable as it helps broaden literature coverage. However, the protocol does not mention how non-English literature will be translated, or how consistency and accuracy in data extraction from these different language sources will be ensured during the data extraction process. A brief explanation of the relevant process or resources could be beneficial. The original timeline was planned at the time of protocol submission (February 20, 2025); however, since the manuscript is still under review, we have revised the text to clearly state that screening will begin only after the protocol is accepted. The estimated durations for each methodological phase (screening, data extraction, synthesis, and manuscript drafting) have been retained but are now referenced relative to the approval date.

We have clarified how non-English literature will be handled. Publications in English, French, Romanian, and Russian will be assessed directly by fluent co-authors. For Spanish and German articles, initial screening and data extraction will be supported by AI-assisted translation tools, followed—if needed—by verification through a second reviewer or language specialist.

This clarification was added to the Types of Sources – Language subsection and is marked in green.

2.5. The exclusion criteria state: "research focusing on dietary supplements or pharmacological compounds that are not applicable to military operational nutrition will not be included."

The judgment of what is "not applicable to military operational nutrition" might be somewhat subjective. It is recommended to provide screeners with more specific defining criteria or examples to ensure consistency in the screening process.

We have revised the criterion to make it precise and reproducible. Studies on supplements, ergogenic aids, or pharmacological compounds are excluded if the intervention is not designed for, tested in, or operationally relevant to active-duty military personnel. Research conducted exclusively in civilian, clinical, or sedentary populations—without a direct connection to military performance—is explicitly excluded. This modification is marked in green in the Inclusion / Exclusion Criteria section.

2.6. Minor Suggestions and Clarifications:

In the "Additional Information" section, under "Data Availability," it states: "N/A - No results are reported" and "No datasets were generated or analysed during the current study." However, the main text (line 138) mentions: "At the time of submission, this scoping review is in the initial data extraction phase." If data extraction has already begun, it could be stated that "no datasets have been finalized or fully analyzed at this stage." A more precise statement could be: "Data extraction is currently underway. No datasets have been finalized or fully analyzed at the time of this protocol submission. All relevant data from this study will be made available upon study completion."

We have corrected the “Data Availability” section: the “N/A” statement has been removed, and we have clarified that no datasets are currently available. De-identified data will be made publicly available in an open-access repository after data extraction and synthesis are completed.

2.7. Detail in the "Concept" Definition:

In the "Inclusion Criteria" under "Concept (C)," when describing smart technologies, it mentions "emerging technologies for real-time tracking of physiological and cognitive responses to nutrition, without direct dietary adjustment." This limiting condition "without direct dietary adjustment" requires all reviewers to have a consistent understanding and application during literature screening. We have revised the sentence to provide a precise definition of smart technologies. The new formulation clarifies that “emerging technologies” refers to wearable biosensors, digital platforms, and AI-based systems that monitor physiological or cognitive markers in real time without intervening in dietary intake during the study period. This makes the criterion clear and reproducible for all reviewers. The modification is highlighted in green in the Inclusion Criteria – Concept (C) section.

---

## [Editor Report · Decision Letter 1]

Monitoring Cognitive Resilience in Military Personnel in Extreme Operational Environments: The Role of Smart Technologies and Nutritional Strategies – A Scoping Review Protocol

PONE-D-25-09172R1

Dear Dr. Siminiuc,

We’re pleased to inform you that your manuscript has been judged scientifically suitable for publication and will be formally accepted for publication once it meets all outstanding technical requirements.

Kind regards,

Rogis Baker, Ph.D

Academic Editor

PLOS ONE
---

## [Editor Report · Acceptance letter]

PONE-D-25-09172R1

PLOS ONE

Dear Dr. Siminiuc,

I'm pleased to inform you that your manuscript has been deemed suitable for publication in PLOS ONE. Congratulations! Your manuscript is now being handed over to our production team.

Kind regards,

on behalf of

Dr. Rogis Baker

Academic Editor

PLOS ONE